# The Combined Potential of PRP and Osteoinductive Carrier Matrices for Bone Regeneration

**DOI:** 10.3390/ijms26178457

**Published:** 2025-08-30

**Authors:** Anastasiia Yurevna Meglei, Irina Alekseevna Nedorubova, Viktoriia Pavlovna Basina, Viktoria Olegovna Chernomyrdina, Dmitry Vadimovich Goldshtein, Tatiana Borisovna Bukharova

**Affiliations:** Research Centre for Medical Genetics, 115478 Moscow, Russia; an.megley95@yandex.ru (A.Y.M.); vika.basina12@gmail.com (V.P.B.); victoria-mok@yandex.ru (V.O.C.); dvgoldshtein@gmail.com (D.V.G.); bukharova-rmt@yandex.ru (T.B.B.)

**Keywords:** PRP, PRF, biomaterials, MSCs, BMP-2, regenerative medicine, bone regeneration

## Abstract

In regenerative medicine, orthobiologics, particularly platelet-rich plasma (PRP), are widely used due to their ability to enhance natural tissue repair mechanisms. PRP contains a concentrated pool of growth factors and cytokines that enhance regeneration while also acting as a biomimetic scaffold, thereby optimizing the microenvironment for tissue healing. In bone tissue engineering, PRP is commonly combined with synthetic or natural biomaterials, as its fibrin matrix alone lacks sufficient mechanical stability. However, even such composite systems frequently exhibit limited osteoinductive capacity, necessitating further supplementation with bioactive components. This review evaluates the regenerative potential of PRP in bone defect healing when combined with osteoinductive agents in preclinical in vivo models. We present compelling experimental evidence supporting the efficacy of this combined therapeutic approach.

## 1. Introduction

In recent decades, regenerative medicine has seen growing interest in approaches aimed at stimulating the body’s intrinsic resources to repair damaged tissues. In this regard, orthobiologic preparations, which consist of autologous components, have emerged as a promising tool for promoting the formation of functional tissues at the injury site [1]. Platelet concentrates hold a special place among orthobiologic preparations, as they not only contain growth factors and cytokines within their granules that enhance regeneration but can also serve as biomimetic scaffolds.

Most platelet concentrate products are broadly termed PRP (Platelet-Rich Plasma), although this generalization does not reflect the specific characteristics of the final product. According to the Dohan Ehrenfest classification, which is based on leukocyte content and fibrin matrix density, four categories are distinguished: platelet-rich plasma with low leukocyte content (P-PRP), platelet-rich plasma with high leukocyte content (L-PRP), platelet-rich fibrin with low leukocyte content (P-PRF), and platelet-rich fibrin with high leukocyte content (L-PRF) [2]. The PRP classification has demonstrated that each category possesses distinct properties and specific potential clinical applications. P-PRP and L-PRP can be used either as liquid injectable solutions or in the form of an activated gel applied to the injured area. In contrast, P-PRF and L-PRF are activated gels that, due to their dense fibrin network, can serve as a matrix and provide sustained release of multiple growth factors over an extended period [3].

Furthermore, fibrin gel obtained by mixing commercial fibrinogen and thrombin has been shown in several studies to be a suitable material for the localized delivery of growth factors to the injury site [4,5].

Objective: This review summarizes current literature data on the effect of PRP in combination with osteoinductive components, such as bone morphogenetic protein 2 (BMP-2), genetic constructs carrying the *BMP2* gene, and stem cells, including genetically modified ones, on the efficacy of bone regeneration in preclinical models.

## 2. Role of PRP in Bone Regeneration

The orthobiological role of PRP in tissue regeneration is to imitate the initial stage of the healing cascade—hematoma formation [6,7,8]. Although regeneration processes vary among different tissues, the formation of a platelet-fibrin clot represents a universal and essential step in initiating the molecular events that ultimately lead to mature tissue formation. Effective bone regeneration requires modulation of three key processes: inflammation regulation, vascular network formation, and osteogenic differentiation of progenitor cells at the defect site [9,10,11,12]. Growth factors and cytokines released from activated platelets are involved in the regulation of inflammation and angiogenesis [8,11,13,14,15]. The interaction between platelets and various leukocyte subsets may exert immunomodulatory effects by influencing macrophage polarization [6,11]. Several studies have demonstrated that PRP administration promotes osteoblastic differentiation of mesenchymal stem cells (MSCs) [16,17]. Furthermore, PRP can stimulate chemokinesis and chemotaxis of osteoblast-like cells through cytoskeletal reorganization, thereby accelerating their migration [18] (Figure 1).

Despite the ability of PRP to enhance cell proliferation, migration, and osteogenic differentiation in vitro [19], conflicting evidence exists regarding its capacity to stimulate bone healing in vivo. PRP administration alone had no significant effect on bone formation in femoral and tibial defect models, with the mineral density of newly formed bone being substantially lower than that of native bone tissue [20,21,22,23]. This suggests that PRP alone may be insufficient to achieve therapeutic efficacy, which could be attributed to two key factors. First, single-dose platelet concentrate administration cannot provide sustained therapeutic effects due to the rapid release and subsequent degradation of growth factors. Second, PRP lacks the structural integrity to withstand mechanical loading [24] and therefore cannot serve as an osteoconductive scaffold. Although PRP shows considerable promise as a potent enhancer of bone regeneration, growing evidence underscores the necessity of combining PRP with various biomaterials and bioactive components.

## 3. Platelet Concentrates Combined with Biomaterials

Current approaches to enhancing bone regeneration are based on combining PRP with biomaterials of various compositions and origins. This combination provides both structural support and may facilitate sustained release of growth factors from PRP, thereby promoting the bone regeneration process.

Skwarcz et al. investigated the efficacy of a bone graft composed of 80% tricalcium phosphate and 20% hydroxyapatite combined with PRP for bone regeneration. Micro-CT imaging performed at 8 and 16 weeks post-implantation demonstrated that supplementing the osteoconductive material with PRP as a growth factor source significantly enhanced bone formation [25]. A number of other studies also highlight the beneficial role of PRP when combined with osteoconductive scaffolds [22,26,27,28]. Several research groups have demonstrated that combining various osteoconductive carriers with PRP is more effective for bone regeneration compared to scaffold-only applications [22,26]. Zhang et al. reported that PRP combined with bioactive borate glass resulted in significantly improved defect closure and higher mineralization density of new formed bone at 12 weeks post-implantation compared to bioactive borate glass alone [27]. Similarly, Bahraminasab et al. found that a 3D PLA/nano-hydroxyapatite matrix combined with PRP showed the most complete defect healing in histological assessments [28].

However, the effect of PRP combined with bone substitutes on regeneration efficacy remains controversial. In contrast to the aforementioned studies, numerous reports have documented no beneficial effect of PRP when added to osteoconductive matrices [15,29,30,31,32,33,34]. Implantation of platelet products combined with collagen sponge [29], hydroxyapatite [31], or beta-tricalcium phosphate (β-TCP) [30] into calvarial defects showed no statistically significant differences in the volume or density of newly formed bone compared to scaffold-only implantation. Any observed effects were limited to early time points, with no differences detected at later stages. In studies using femoral and tibial bone models, scaffolds incorporating PRP with chitosan/hydroxyapatite [32], PRP with alginate [34], PRF-BioOss [33], or PRP-gelatin methacrylate [15] demonstrated enhanced bone regeneration. However, these studies either lacked comparisons between the individual use of biomaterials, which does not allow for the effect of combined or isolated use to be judged, or showed no statistical differences between experimental groups.

Clinical trial data demonstrate ambiguous effects of PRP application in dental patients. In bone augmentation procedures, such as sinus grafting of severe maxillary atrophy, the addition of PRF to the conventionally used deproteinized bovine bone (Bio-Oss) significantly reduced the healing time favoring optimal bone regeneration [35]. Allografts mixed with PRP enhanced bone regeneration in patients with ridge defects [36]. However, randomized controlled clinical trials have reported mixed outcomes: only one study found a significant difference in bone augmentation with PRP supplementation during sinus grafting procedures, while four others showed no statistically significant improvement compared to non-PRP control groups [37].

Platelet concentrates demonstrated efficacy when combined with autografts for bone formation and maturation [38,39]. However, comparative analysis of PRP-autograft versus PRP with other bone graft materials reveals that the regenerative effects are primarily attributable to the autograft properties, as a natural source of osteoprogenitor cells, osteoinductive factors, and growth factors, rather than to PRP’s contribution [40,41].

Emerging experimental evidence suggests that novel approaches should focus on enhancing the osteoinductive properties of engineered PRP-containing scaffolds

## 4. Platelet Concentrates Combined with BMP-2

One promising approach to enhance the osteoinductive properties of PRP-containing scaffolds involves their combination with osteoinductive proteins (Table 1). Members of the BMP protein family represent particularly potent osteogenic inducers and are among the most frequently employed agents for bone regeneration. Studies by Berner et al. and Jung-Hyun Park et al. demonstrated that calcium phosphate-coated nanofiber mesh tube loaded with PRP/BMP-7 combinations [42], or L-PRF combined with BMP-2-soaked collagen sponges [43], respectively, exhibited significant regenerative potential. However, Elsalanty et al. reported that supplementing demineralized bone matrix with PRP did not significantly improve bone regeneration outcomes, even with rhBMP-2 administration [44].

Most existing studies focus on evaluating the efficacy of autologous platelet concentrates combined with scaffolds and BMPs. In a comparative study, Liu et al. demonstrated that incorporating a blood clot into calcium phosphate scaffolds impregnated with BMP-2 facilitated sustained growth factor release. This combination exhibited osteogenic effects in both ectopic and orthotopic osteogenesis models. Notably, when implanted into rat calvarial defects, scaffolds containing BMP-2 and blood clots promoted mature bone formation by week 12, unlike control groups without the blood component [45]. The same research team conducted a pilot clinical trial confirming the regenerative potential of these scaffolds, with complete defect healing observed within six months following implantation in tibial or humeral bone defects [45]. To further enhance the regenerative potential of osteoinductive scaffolds, platelet concentrates can be modified, as demonstrated by Park et al. Their study examined polyglycolic acid-based scaffolds coated with fibronectin and supplemented with modified PRP enriched with angiogenic factors, combined with BMP-2. In vitro evaluation revealed significant stimulation of endothelial cell migration, while in vivo implantation of the composite scaffold in rat calvarial defects significantly increased vascularization and showed enhance bone formation compared to scaffolds containing either BMP-2 or PRP alone, as confirmed by histological and micro-CT analysis [47]. Notably, the combination of modified PRP with BMP-2 allowed for reduced BMP-2 dosage while maintaining therapeutic efficacy.

Several studies have demonstrated the efficacy of commercial fibrin gels as delivery matrices for osteoinductive proteins [4,5,46]. In a murine femoral defect model, Chen et al. showed that BMP-2-loaded fibrin matrices promoted complete defect healing with mature bone formation, as confirmed by histological and micro-CT analysis, outperforming direct protein injection alone [5]. Similarly, Kaipel et al. reported significant bone regeneration when combining BMP-2 with fibrin gel in femoral defects [46]. Complementary findings by Li et al. demonstrated that in a rabbit tibial defect model, fibrin matrices incorporating BMP-7 resulted in significantly higher mineral density and greater bone volume compared to direct BMP-7 administration without a fibrin carrier [4].

Current evidence strongly supports the therapeutic advantages of fibrin-based materials or PRP when combined with BMPs. Importantly, well-designed comparative studies incorporating appropriate non-fibrin control groups [4,5,45,47] allow differentiating the contribution of platelet concentrates and osteoinductive components and highlighting the clinical value of this combinatorial strategy.

## 5. Platelet Concentrates Combined with Gene Vectors

As a promising alternative to enhance the osteoinductive properties of PRP-based scaffolds, numerous studies have employed genetic constructs encoding osteoinductive factors for localized therapy (Table 2). The incorporation of genetic vectors into the defect site enables controlled and sustained production of therapeutic proteins by both resident and recruited cells. PRP supplementation may further augment the migration of osteoprogenitor cells through its endogenous growth factor content.

To confer osteoinductive properties to polylactide (PLA) granule-based scaffolds with PRP, adenoviral vectors carrying the *BMP2* gene were employed [48]. The PRP hydrogel facilitated gradual release of genetic constructs by forming a clot containing PLA particles and viral vectors. Although experimental data for a PRP-free control group were not provided, the presented results strongly support the promise of this combined approach.

Commercial fibrin gels are also employed for delivering genetic constructs to bone defect sites. Implantation of fibrin gel combined with plasmids carrying *BMP2/BMP7* genes into femoral bone defects showed a trend toward increased bone volume in the pBMP2/7 group compared to fibrin gel alone [46]. In the study by Behnoush et al., a multicomponent fibrin-based matrix incorporating bioactive components, combined with a gene-activated collagen matrix containing BMP2/FGF2 plasmid vectors, stimulated ectopic bone formation compared to control groups [51]. While fibrin gel shows potential as a carrier for genetic constructs, its efficacy in these studies was not conclusively demonstrated due to methodological limitations, particularly the lack of control groups with alternative carriers or complete absence of fibrin components.

Several studies have investigated the efficacy of combining natural or synthetic polymers with PRP containing plasmid constructs carrying the *BMP2* gene [49,50]. Incorporating PRP into gene-activated scaffolds significantly enhanced their osteogenic potential through sustained release of genetic constructs. PLA granule-based scaffolds combined with PRP and polyplexes containing the *BMP2* gene demonstrated more pronounced osteogenic differentiation of MSCs and improved healing of rat calvarial defects compared to PRP-free gene-activated scaffolds [50]. Adding plasmids with *BMP2* gene to collagen and PRP-based scaffolds enhanced cell survival and upregulated osteogenic differentiation genes. In vivo experiments further revealed that PRP incorporation doubled the volume of newly formed bone compared to PRP-free scaffolds [49]. These findings confirm PRP’s critical role in accelerating bone tissue repair.

Moreover, data from Nedorubova et al. and Meglei et al. highlight the necessity of including control groups without fibrin-containing components and demonstrate that composite scaffolds combining platelet concentrates with genetic constructs encoding osteoinductive factors outperform biomaterials lacking platelet-derived components in stimulating neo-osteogenesis.

## 6. Platelet Concentrates Combined with Stem Cells

MSCs promote bone tissue regeneration by differentiating into osteoblasts and stimulating new bone matrix formation. However, delivery of isolated MSCs to the defect site often demonstrates limited efficacy due to rapid cell elimination by tissue fluid and migratory cell loss, which reduces therapeutic efficacy. The use of cytocompatible scaffolds seeded with cells as part of a complex biotransplant represents a promising approach for bone regeneration (Table 3).

Studies by Almansoori et al. and Park et al. demonstrated that both synthetic and natural bone substitutes exhibit more pronounced regenerative effects when combined with platelet concentrates and cells [54,55]. Wang et al. showed that platelet gel maintained the viability of adipose derived stem cells (ADSCs) encapsulated in a three-dimensional porous titanium scaffold. Twelve weeks after implantation into rabbit femoral bone defects, this composite stimulated cell migration and differentiation at the injury site. Moreover, growth factors within the PRP-incorporated matrix promoted vascular network formation, thereby accelerating bone tissue regeneration [52]. Notably, newly formed bone tissue was observed not only at the scaffold-host bone interface but also within the scaffold’s central regions, confirming the osteoinductive properties of PRP-containing scaffolds. The control group without PRP showed no significant regenerative effects.

The combination of autologous bone particles with PRP and bone marrow stromal cells (BMSCs) also enhances bone regeneration, as evidenced in a rabbit radial defect model through radiographic, histological, and histomorphometric analyses. The PRP-supplemented implant group demonstrated superior bone formation compared to PRP-free implants [19]. According to the authors, this outcome stems from PRP-enhanced BMSCs proliferation via platelet-derived growth factor release, aligning with Tajima et al. findings in a rat calvarial defect model. Their work proved that specifically the ADSC/PRP combination increased growth factor production and osteogenic cell populations, markedly stimulating osteogenesis—an effect absents in both PRP-free groups and alternative carrier groups [53]. This confirms that transplanted cells struggle to maintain activity without appropriate carriers and growth factors, directly compromising therapeutic efficacy.

Therefore, controlled studies with PRP-free groups [19,52,53] provide robust evidence that platelet concentrates significantly amplify the regenerative potential of cell-seeded biomaterials by improving the survival of implanted cells, stimulating angiogenesis, and promoting more active bone formation within defects.

## 7. Platelet Concentrates Combined with Gene-Modified Stem Cells

An alternative approach to enhance osteoinductive properties may involve using pre-modified MSCs in combination with PRP-based biomaterials (Table 4).

The study by Bukharova et al. demonstrated that PRP and bone chip-based carrier scaffolds “Osteomatrix” supported the survival of genetically modified cells producing BMP-2, which could potentially enhance regenerative processes in the bone defect area [58]. Since some literature reports indicate the low efficacy of PRP due to the rapid release and degradation of cytokines and other growth factors essential for bone regeneration, Fernandes et al. proposed encapsulating PRP in alginate granules to enable gradual release of platelet concentrates. They used them in combination with MSCs transduced with adenoviral vectors carrying the *BMP2* gene, which significantly enhanced osteogenic differentiation in vitro [57]. In a later study, Bukharova et al. described the ability of PLA and PRP-based scaffolds with Ad-BMP2-transduced MSCs to stimulate osteogenesis in rat calvarial defects. Their implantation resulted in twice the area of newly formed bone tissue compared to the group where scaffolds impregnated with viral vectors carrying the *BMP2* gene were used [48]. Zunpeng Liu et al. demonstrated that the combination of PRP and Ad-BMP2-transduced MSCs within a nano-calcium sulfate based matrix significantly increased the amount of newly formed bone compared to both PRP-free scaffolds and those containing unmodified MSCs, as evidenced by histological and micro-CT analyses [56].

Thus, preliminary modification of cells encapsulated in a PRP-based scaffold may represent another effective approach in bone regeneration, providing a pool of genetically programmed osteoblast-directed cells that stimulate the natural cascade of regenerative processes.

## 8. Conclusions

PRP-based fibrin gel has demonstrated efficacy in numerous studies due to its biocompatibility, ability to support cell proliferation and adhesion. The possibility of autologous production of PRP makes materials based on it promising for clinical use in regenerative medicine in terms of safety and personalized approach. However, clinical trial data indicate an ambiguous effect of adding PRP alone to standard bone grafting materials in dentistry, underscoring the need for developing an improved strategy.

This review provides a detailed analysis of studies investigating the combined use of PRP and fibrin gels with osteoinductive components, such as osteoinductive proteins, genetic constructs carrying their genes, and modified MSCs. Particular attention is paid to comparative studies with control groups without the introduction of fibrin-containing components, which made it possible to evaluate the contribution of PRP to the process of bone regeneration. These studies demonstrated that addition of PRP to osteoinductive materials enhances their effect, promoting faster and more effective healing of bone defects compared to groups not treated with platelet concentrates.

Furthermore, as shown by some researchers, additional modification of PRP by enriching it with angiogenic factors or fibrin gels by incorporating nanoparticles that enhance the interaction between inducers and their receptors can further amplify the osteogenic effect of biomaterials for bone regeneration.

Thus, the obtained results provide compelling evidence supporting the use of easily accessible patient-derived PRP in dental or surgical practice to enhance the efficacy of osteoplastic materials combined with osteoinductive agents. PRP-based biomaterials could significantly advance the treatment of bone tissue defects, including extensive bone loss and fractures with delayed union. However, widespread clinical adoption requires protocol standardization, further preclinical studies, including not only experimental PRP groups but also control groups without PRP or with alternative carriers, as well as clinical trials confirming the safety and efficacy of such approaches.

## Figures and Tables

**Figure 1 ijms-26-08457-f001:**
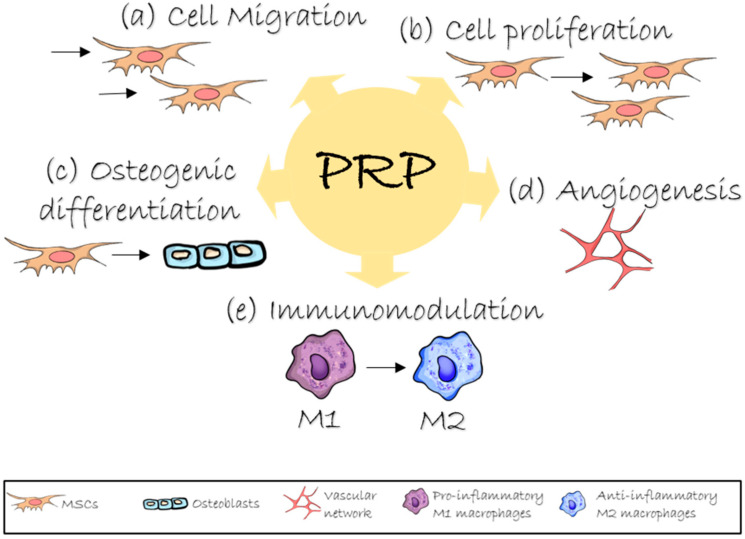
Role of PRP in bone regeneration. PRP’s growth factors act to: (**a**) stimulate migration and (**b**) proliferation of stem and progenitor cells to the injury site, (**c**) promote osteogenic differentiation into bone-forming osteoblasts, (**d**) induce angiogenesis to create new blood vessels and (**e**) modulate immunity by shifting macrophages from a pro-inflammatory (M1) to an anti-inflammatory (M2) phenotype.

**Table 1 ijms-26-08457-t001:** Combined application of scaffolds with platelet concentrates and inductive protein supplementation.

Biomaterial	Model/Defect	Osteoinductor	Results	Reference
Calcium phosphate cement, blood clot and dexamethasone	Rats/Calvarial defect; Patients/Tibial plateau fractures or proximal humeral fractures	rhBMP-2	The addition of a blood clot to the scaffold resulted in more efficient bone formation compared to the scaffold alone. The introduction of dexamethasone effectively facilitated M2 polarization of macrophages	[45]
Fibrin gel incorporated with sulfated chitosan nanoparticles	Mice/Femoral defect	rhBMP-2	Fibrin scaffold combined with an inductive protein demonstrated the highest bone formation when compared to empty defects or protein-only administration. Sulfated chitosan nanoparticles regulate the macrophage polarization	[5]
Fibrin gel	Rabbits/Tibial defect	rhBMP-2	The mineral density and new bone volume were significantly higher in the rhBMP-2 + fibrin gel group compared to other treatment groups	[4]
Fibrin gel	Rats/Femoral defect	rhBMP-2	In the group using fibrin gel in combination with protein, a trend towards increased bone volume was observed compared to fibrin implantation alone	[46]
Calcium phosphate-coated nanofiber mesh tube and PRP	Rats/Femoral defect	BMP-7	Scaffolds containing BMP-7 and PRP resulted in complete defect closure by 12 weeks after implantation compared to both empty scaffolds and PRP only scaffolds	[42]
Fibronectin-coated PGA scaffolds and modified PRP	Rats/Calvarial defect	rhBMP-2	PGA scaffolds combined with both PRP and rhBMP-2 promoted more substantial bone defect healing than either PGA+PRP or PGA+rhBMP-2 alone. Modified PRP induced faster migration of cord blood-derived outgrowth endothelial-like cells and significantly increased numbers of blood vessels	[47]
L-PRF and collagen sponge	Patients/Medication-related osteonecrosis of the jaws	rhBMP-2	The combined application of BMP-2 and L-PRF led to earlier new bone formation relative to L-PRF administration alone	[43]
Demineralized bone matrix and PRP	Dogs/Calvarial defect	rhBMP-2	PRP supplementation showed no significant effect on the regeneration process, regardless of rhBMP-2 co-administration	[44]

**Table 2 ijms-26-08457-t002:** Combined application of scaffolds with platelet concentrates and genetic constructs.

Biomaterial	Model/Defect	Osteoinductor	Results	Reference
Polylactide particles PRP-based fibrin clot	Rats/Calvarial defect	Ad-BMP2	The gene-activated PLA/PRP-Ad-BMP2 scaffolds promoted more substantial new bone formation compared to empty defects	[48]
Collagen-based scaffolds with PRP	Rats/Calvarial defect	pBMP2	Incorporation of PRP into gene-activated scaffolds resulted in a two-fold increase in newly formed bone volume compared to PRP-free scaffolds	[49]
Polylactide particles and PRP-based fibrin clot	Rats/Calvarial defect	pBMP2	Gene-activated scaffolds incorporating PRP demonstrated enhanced osteogenic differentiation of MSCs and improved healing of critical-sized rat calvarial defects compared to PRP-free gene-activated carriers	[50]
PLGA-microparticles embedded in a fibrin gel surrounded by a collagen matrix	Rats/Ectopic osteogenesis	pBMP-2	The gel-containing material stimulated ectopic bone formation compared to other experimental groups	[51]
Fibrin gel	Rats/Femoral defect	pBMP2/7	The fibrin gel with pBMP2/7 combination showed a trend toward increased bone volume compared to fibrin-only implants	[46]

**Table 3 ijms-26-08457-t003:** Combined application of scaffolds with platelet concentrates and stem cell supplementation.

Biomaterial	Model/Defect	Osteoinductor	Results	Reference
3D-porous titanium alloy implants and PRP	Rabbits/Femoral defect	ADSC	Both bone volume and mineral density were significantly higher in the group receiving titanium scaffolds combined with PRP and cells, compared to control groups containing either scaffolds alone or cell-seeded scaffolds. The introduction of PRP significantly increased tube formation, expression of angiogenic markers and CD31+ cells in the defect site	[52]
PRP	Rats/Calvarial defect	ADSC	The most pronounced bone tissue regeneration was observed following ADSCs/PRP implantation compared to ADSCs/col1, PRP-only, or type 1 collagen-only groups	[53]
Polycaprolactone-β tricalcium phosphate bio-scaffold and PRP	Pigs/Mandibular defect	MSC	Implantation of PCL-TCP + MSCs + PRP scaffolds resulted in significantly greater bone formation area and higher mineral density compared to unseeded scaffold controls	[54]
Allogenic bone graft and PRP	Rabbits/Femoral defect	BMSC	The combination of MSCs and PRP enhanced the expression of osteogenic markers compared to the cell-free control group	[55]
Autogenous bone particles and PRP	Rabbits/Radial diaphysis defect	BMSC	The newly formed bone fraction area was significantly larger in the group receiving autologous bone chips combined with PRP and cells compared to the group without PRP administration	[19]

**Table 4 ijms-26-08457-t004:** Combined application of scaffolds with platelet concentrates and genetically modified stem cells.

Biomaterial	Model/Defect	Osteoinductor	Results	Reference
Nano-calcium sulfate and PRP	Rats/Calvarial defect	Ad-BMP2- MSC	The combination of BMP2-modified MSCs with nCS/PRP scaffolds resulted in increased volume and mineral density of regenerated bone compared to control groups lacking either PRP or BMP2-modified MSCs	[56]
PRP with alginate microspheres	in vitro	Ad-BMP2- MSC	The osteogenic differentiation efficiency was significantly higher in the MSC/BMP2 + PRP group compared to the non-PRP control group	[57]
Polylactide particles and PRP-based fibrin clot	Rats/Calvarial defect	Ad-BMP2- MSC	Scaffolds containing transduced MSCs demonstrated faster and more pronounced bone defect regeneration compared to empty defects	[48]
Osteomatrix and PRP	in vitro	Ad-BMP2- MSC	The introduction of platelet-rich plasma (PRP) into tissue engineering scaffolds improves cell distribution and viability, while augmenting BMP-2 protein secretion	[58]

## Data Availability

Data are contained within the article.

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
