# Peer review of "The Combined Potential of PRP and Osteoinductive Carrier Matrices for Bone Regeneration"

_ijms, 2025, doi:10.3390/ijms26178457_

Round 1
Reviewer 1 Report
Comments and Suggestions for Authors
The review manuscript focuses on summarizing the regenerative potential of PRP in bone defect healing. It is a well-written manuscript with good organization. The paper covers the major areas related to the application of PRP in bone defect repair and regeneration. The topic is original and important to the field of skeletal tissue engineering. The conclusion effectively reflects the key points presented in the manuscript. This review will benefit a broad readership engaged in research on skeletal tissue engineering and regenerative medicine. However, the manuscript should be further improved before being considered for publication.
1. The drawings of Figures 1 and 2 could be further improved. In addition, a more detailed description should be provided for each figure legend.
2. The most recent references related to PRP research in bone defect repair and regeneration should be incorporated into each section.
3. More in-depth discussion in each section would provide valuable insights for the readership’s research. For example, in Section 4, the regenerative efficacy of PRP combined with BMPs may be dose-dependent. A high dosage of BMPs may cause side effects. What is the benefit of using a combination of PRP and BMPs rather than BMPs alone? The advantages and side effects of PRP and BMPs should be discussed in greater depth. In Section 5, a BMP2 plasmid vector was used in combination with PRP to enhance bone regeneration. Why was the BMP2 plasmid vector chosen? Is it more effective than applying the BMP2 growth factor directly? In Section 6, the lack of discussion regarding the current challenges of using PRP with stem cells alone should be addressed. This provides the rationale for Section 7, where gene-modified stem cells are further studied.
Author Response
We would like to thank reviewer for the useful comments and suggestions on our manuscript ijms-3791644, entitled “The Combined Potential of PRP and Osteoinductive Carrier Matrices for Bone Regeneration”. We have provided detailed responses to each of your points below and made the necessary changes based on your recommendations.
- The drawings of Figures 1 and 2 could be further improved. In addition, a more detailed description should be provided for each figure legend.
Response 1: In response to your suggestion, we have improved Figure 1. We have also expanded its legend to provide a more detailed description of the presented elements and processes. Figure 2 has been removed from the manuscript. This decision was made in direct response to a comment from another reviewer, who rightly pointed out that the figure was redundant. We agree that removing it from the manuscript helps to eliminate repetition.
- The most recent references related to PRP research in bone defect repair and regeneration should be incorporated into each section.
Response 2: In direct response to your comment, we have performed an extensive new literature search, focusing specifically on studies published within the last 2-3 years. After this thorough examination, we have confirmed that the current reference list already includes the key recent works in this specific niche of PRP research combined with osteoinductive factors for bone repair.
We sincerely appreciate the reviewer's vigilance on this matter. Should you be aware of any specific, seminal recent publications that you feel we have inadvertently overlooked, we would be very grateful if you could share these citations. We are fully prepared to incorporate any such vital studies to further strengthen the manuscript.
- More in-depth discussion in each section would provide valuable insights for the readership’s research. For example, in Section 4, the regenerative efficacy of PRP combined with BMPs may be dose-dependent. A high dosage of BMPs may cause side effects. What is the benefit of using a combination of PRP and BMPs rather than BMPs alone? The advantages and side effects of PRP and BMPs should be discussed in greater depth. In Section 5, a BMP2 plasmid vector was used in combination with PRP to enhance bone regeneration. Why was the BMP2 plasmid vector chosen? Is it more effective than applying the BMP2 growth factor directly? In Section 6, the lack of discussion regarding the current challenges of using PRP with stem cells alone should be addressed. This provides the rationale for Section 7, where gene-modified stem cells are further studied.
Response 3: We sincerely thank you for this insightful comment and for highlighting these fascinating aspects of the field. We agree that discussions on dose-dependency of BMPs, vector selection, and the specific challenges of stem cell therapy are of great importance for the research community.
The primary aim of our review was to provide a comprehensive descriptive overview of the diverse strategies employed in the literature, mapping the entire landscape of how PRP is combined with various osteoinductive components to enhance bone regeneration. Consequently, our analytical focus was deliberately placed on the role and contribution of PRP within these combinatorial strategies, rather than on a comparative analysis of the advantages and disadvantages of each individual component (e.g., BMP-2 vs. BMP-7, plasmids vs. viral vectors), which itself is a vast and complex topic.
In response to the feedback received and to better reflect the primary focus of our study, we have refined the formulation of the review's objective.
Reviewer 2 Report
Comments and Suggestions for Authors
To whom it may concern:
The following comments concern the manuscript “The Combined Potential of PRP and Osteoinductive Carrier Matrices for Bone Regeneration” prepared by Anastasiia Yurevna Meglei et al., coded as ijms-3791644. I appreciate the authors’ efforts in compiling and organizing a substantial body of literature on this important topic. The suggestions listed in pdf file are intended to help further strengthen the clarity, depth, and impact of the work prior to publication.

Author Response
We would like to thank reviewer for the useful comments and suggestions on our manuscript ijms-3791644, entitled “The Combined Potential of PRP and Osteoinductive Carrier Matrices for Bone Regeneration”. Your insightful suggestions have helped us improve the quality of our work. We have checked and revised the manuscript according to the comments and suggestions, and the detailed responses have been listed below with all the changes and improvements included in the revised manuscript.
- The PDF file of the manuscript contains several yellow-highlighted passages. While the purpose of these highlights is unclear, they do not prevent a full review of the manuscript.
Response 1: The yellow highlights in the PDF file were added by the journal editors during the review stage by the scientific editor. They are intended to visually indicate the edits we made and to simplify the process of checking their correctness. We have now removed highlights from the manuscript. All subsequent changes are properly tracked using the 'Track Changes' mode for clear review.
- Please consider including additional references to other review papers. Given the large body of literature on “PRP Bone Regeneration,” a brief discussion of the contributions of other key reviews – following the basic background and preceding the stated objectives – would help situate this manuscript within the existing body of knowledge and enhance its perceived value.
Response 2: In preparing the manuscript, we incorporated reviews on the general role of PRP in bone regeneration within the appropriate section. Our review focuses on the combined use of PRP and osteoinductive factors. Currently, there are no specialized review articles on this topic to our knowledge. Therefore, our analysis was based primarily on a comprehensive study of the available experimental studies in order to create an original review in this area. We would be very grateful if you could recommend any specific review articles that you think would be particularly appropriate to cite in this context. We are ready to incorporate such suggestions to further improve the manuscript.
In response to the feedback received and to better reflect the primary focus of our study, we have refined the formulation of the review's objective.
- The review tables could benefit from more in-depth analysis. The four tables in the manuscript effectively summarize the reviewed literature, with the “Result” column presenting key findings, some of which are also addressed in the text. However, to enrich the review, it would be useful to indicate, for example, whether the bone regeneration mechanisms discussed focus more on inflammation regulation, vascular network formation, or osteogenic differentiation, and whether the materials employed involve unique fabrication methods or formulations. If adding this level of detail to the tables would make them overly dense, consider incorporating more discussion in the main text.
Response 3: We have thoroughly re-examined the literature included in the tables. Most studies have focused on the investigation of osteogenic differentiation and bone formation with the combined use of PRP and osteoinductive components. A few number of studies investigated effects on alternative regenerative pathways, such as angiogenesis or immunomodulation. In accordance with your recommendation, we have updated the "Results" column in the tables for those cases where data on angiogenesis or immunomodulation was reported.
- The necessity of Figure 2 may be reconsidered. At present, it appears to present only three interconnected concepts, which could be conveyed effectively in text. Including additional mechanisms or interactions could enhance the figure’s significance.
Response 4: In response to your comment, we have removed Figure 2. We believe this change improves the flow of the manuscript and eliminates redundant visual material.
- In lines 218 and 258, the use of the term “Thus” does not seem to match the flow and meaning of the surrounding context. Alternative transition words may better convey the intended logic.
Response 5: We agree that the use of "Thus" in the highlighted instances did not accurately reflect the intended logical relationships between the ideas. In direct response to your suggestion, we have replaced inappropriate uses of "Thus" with more precise transition words, such as "Moreover," and "Therefore," depending on the context. We agree that these changes markedly improve the logical coherence of the work.
- As a minor suggestion, it is generally preferable to avoid citing references in the conclusion. The citations [47] in line 306 and [5] in line 307 could be removed, though this is optional.
Response 6: In accordance with your feedback, we have removed references [47] and [5] from lines 306 and 307, respectively. We agree that the conclusion should present a synthesized summary of the study's findings without direct citations.